# Human Papillomavirus Infection and the Risk of Erectile Dysfunction: A Nationwide Population-Based Matched Cohort Study

**DOI:** 10.3390/jpm12050699

**Published:** 2022-04-27

**Authors:** Sin-Ei Juang, Kevin Sheng-Kai Ma, Pei-En Kao, James Cheng-Chung Wei, Hei-Tung Yip, Mei-Chia Chou, Yao-Min Hung, Ning-Chien Chin

**Affiliations:** 1Department of Anesthesiology, Kaohsiung Chang Gung Memorial Hospital, Kaohsiung 833401, Taiwan; juangsinei@gmail.com; 2Center for Global Health, Perelman School of Medicine, University of Pennsylvania, Philadelphia, PA 19104, USA; sheng.kai.ma@cern.ch; 3Department of Epidemiology, Harvard T.H. Chan School of Public Health, Boston, MA 02115-5810, USA; 4Graduate Institute of Biomedical Electronics and Bioinformatics, National Taiwan University, Taipei 11114, Taiwan; 5School of Medicine, Chung Shan Medical University, Taichung 40402, Taiwan; kaoanson98@gmail.com; 6Institute of Medicine, Chung Shan Medical University, Taichung 40402, Taiwan; wei3228@gmail.com; 7Graduate Institute of Integrated Medicine, China Medical University, Taichung 40402, Taiwan; 8Division of Allergy, Immunology and Rheumatology, Chung Shan Medical University, Taichung 40402, Taiwan; 9Department of Management office for Health Data, China Medical University Hospital, Taichung 40402, Taiwan; fionyip0i0@gmail.com; 10College of Medicine, China Medical University, Taichung 40402, Taiwan; 11Institute of Public Health (Biostatistics), National Yangming University, Taipei 112304, Taiwan; 12Department of Recreation and Sports Management, Tajen University, Pingtung County 907101, Taiwan; meeichia@gmail.com; 13Department of Physical Medicine and Rehabilitation, Kaohsiung Veterans General Hospital, Pingtung Branch, Pingtung County 907101, Taiwan; 14Department of Internal Medicine, Kaohsiung Municipal United Hospital, Kaohsiung 813414, Taiwan; 15Shu-Zen Junior College of Medicine and Management, Kaohsiung 813414, Taiwan; 16Department of Orthopedics, Taichung Veterans General Hospital, Taichung 40402, Taiwan

**Keywords:** HPV, erectile dysfunction, human papillomavirus, cohort study

## Abstract

Background: Male patients with genital warts are known for higher rates of sexual dysfunction. This study was conducted to investigate whether human papillomaviruses (HPV) infection is associated with an increased risk of erectile dysfunction (ED). Methods: Patients aged over 18 with HPV infection (*n* = 13,296) and propensity score-matched controls (*n* = 53,184) were recruited from the Longitudinal Health Insurance Database (LHID). The primary endpoint was the diagnosis of ED. Chi-square tests were used to analyze the distribution of demographic characteristics. The Cox proportional hazards regression was used to estimate the hazard ratios (HRs) and 95% confidence intervals (CIs) for the development of ED in both groups, after adjusting for sex, age, relevant comorbidities, co-medication, and surgery. Results: ED developed in 181 patients of the study group. The incidence density of ED was 2.53 per 1000 person-years for the HPV group and 1.51 per 1000 person-years for the non-HPV group, with an adjusted HR (95% CI) of 1.63 (1.37–1.94). In stratification analysis, adjusted HR of diabetes-, chronic obstructive pulmonary disease (COPD-), and stroke-subgroup were 2.39, 2.51, and 4.82, with significant *p* values for interaction, respectively. Sensitivity analysis yields consistent findings. Conclusions: The patients with HPV infection had a higher risk of subsequent ED in comparison to the non-HPV controls. The mechanism behind such association and its possible role in ED prevention deserves further study in the future.

## 1. Introduction

Human papillomaviruses (HPVs) are small, double-stranded DNA viruses that infect the cutaneous and mucosa epithelium and are the most common sexually transmitted disease in women [1,2]. There are more than 200 recognized types of HPV [3], of which 40 types infect the genital area and a dozen or so types of HPV are known to be high risk and lead to cervical malignancy [4,5,6]. With the administration of HPV vaccines, a significant decrease in the prevalence of HPV infections, cervical lesions, and cervical malignancies in young females has been observed [7,8,9]. Despite the enhanced efficacy, proven safety, and recommendation by the World Health Organization [10], the public’s acceptance of HPV vaccines could still be improved, especially in men [11]. In the U.S. population-based study, the prevalence of genital HPV infection was estimated at 41.3% to 49.3% in males but HPV vaccination coverage was only approximated 7.8% to 14.6% among vaccine-eligible men [12]. Thus, the adverse impact of HPV infections on public health remains. Apart from malignancies, complications of HPV infections include genital warts, bladder cancer, and male infertility due to impaired sperm function [13,14]. However, compared to the extensive number of studies on HPV infections in women, there is a lack of knowledge on HPV infections in male patients [15].

Evidence suggests that males with genital warts have higher rates of sexual dysfunction [16]. Among sexual dysfunction, erectile dysfunction (ED), defined as the inability to maintain or achieve an erection necessary for sexual intercourse, afflicts an estimated 10–20 million men in the United States alone and a survey in Taiwan showed that the prevalence of ED was 27% among all respondents aged not less than 30 years [17,18,19]. The pathogenesis of ED depends on the vascular integrity of the erectile tissue and the HPV infections may correlate with vascular complications [20]. For instance, a case of isolating necrotizing granulomatous vasculitis of the genital area associated with HPV infection has been reported [21]. There are several potential mechanisms underlying the relationship between HPV and ED. First of all, ED is a neurovascular procession that depends on the vascular health of erectile tissue and the nervous system [20]. HPV infection may induce inflammation and cause atherosclerosis progress, which has been shown to seriously affect the vascular health of the erectile tissue, thus contributing to ED pathogenesis [22,23,24,25,26]. Common risk factors for atherosclerosis have been frequently found in patients with ED, with the extent of ED being associated with the number and severity of CVDs [22,23,24,25,26]. For instance, ED has been reported to be associated with elevated high-sensitivity C-reactive protein values and to be correlated with the magnitude of flow-mediated dilation of the brachial artery [23], with the latter being an indicator of endothelial dysfunction [23]. In addition, it has been reported that ED is correlated with coronary calcification and atherosclerotic plaque [23,26] and usually precedes the development of clinically evident CVDs [22,23,26]. These findings indicate that ED may share common pathophysiological mechanisms with CVDs, particularly coronary artery disease (CAD). For instance, both ED and CAD have endothelial dysfunction as part of their pathophysiology [27,28,29], which is evidenced by studies indicating that ED is associated with increased circulating levels of systemic inflammatory [30,31,32,33,34] and endothelial–prothrombotic markers/mediators [35]. As the diameters of the cavernosal arteries are smaller than large coronary arteries, they may be more sensitive to atherosclerosis [36]. Accordingly, HPV-associated atherosclerosis and affected vascular function may be involved in ED development.

Previous studies propose that inflammation caused by viral [37,38] or bacterial [39,40,41,42,43,44] pathogens plays an important role in atherosclerosis pathogenesis. Specifically, previous studies have shown that herpes zoster virus, *Chlamydia pneumoniae*, cytomegalovirus, and hepatitis A virus infection can increase the risk of CAD, myocardial infarction, and even cardiovascular death [45,46,47,48,49]. For instance, previous studies using data from the National Health and Nutrition Examination Survey (NHANES) demonstrated that HPV infection is associated with CVD among women [50]. In addition, infectious agents, including viruses, bacteria, and parasites, can trigger autoimmunity and are associated with secondary vasculitis [51], which may lead to cases such as HPV-associated granulomatous vasculitis of the genital area [21]. The etiologies of ED have also been reported to include chronic systemic vasculitis and severe venous leak caused by venous thrombosis, leading to the fact that HPV infection is related to vascular dysfunction [52,53,54].

Given the fact that genital warts negatively affect male sexual function [16], the manner in which HPV infections fit into our knowledge of ED etiologies remains undetermined. Therefore, we conducted this nationwide population-based cohort study using a longitudinal database in Taiwan to provide epidemiological evidence about the impact of HPV infection on ED development.

## 2. Materials and Methods

### 2.1. Data Source

This study was constructed using longitudinal data from the National Health Insurance Research Database (NHIRD) in Taiwan. The NHIRD contains registry and original claimed data, such as the diagnoses coding, out-patient visits, hospitalization records, medication, and personal information of over 99% of Taiwan’s population. Quarterly expert reviews on randomly sampled claimed data, with a sampling rate of 1/50–100, were performed by the Bureau of National Health Insurance (BNHI). The BNHI also randomly reviews the medical charts of 1/100 ambulatory and 1/20 inpatient claims. This study was approved by the Institutional Review Board of the China Medical University Hospital Research Ethics Committee (CMUH104-REC2-115(AR-4)).

### 2.2. Study Population

Participants with medical records of HPV infection (ICD-9 codes 079.4, 078.10, 078.11, 078.12, 078.19, 795.05, 795.09, 795.15, 795.19, 796.75, and 796.79) between 2000 and 2012 constituted the HPV group (Figure 1). The index date was defined as the date of diagnosis and coding for HPV infection. The control group was selected from the same dataset and was matched by age, sex, index date, co-morbidities, and co-medication at a ratio of 1:4. Individuals diagnosed with ED before the index date or younger than 18 years old were excluded. To ensure the validity of the diagnoses of HPV, patients diagnosed with an HPV infection who never received HPV-associated treatment within three months after the index date were excluded. These treatments included excision, electrocauterization, carbon dioxide (CO_2_) laser operation, or chemosurgery for condyloma, as well as simple or complicated electrocauterization, liquid nitrogen cryosurgery, or cryotherapy using CO_2_ freezing or liquid nitrogen. All participants were followed up until the first medical record of ED or censored on the date of death, or the end of the study (31 December 2013). Finally, 13,296 subjects were included in the HPV group and 53,184 non-HPV subjects served as the control group.

### 2.3. Definitions of the Outcomes and Covariates

The primary endpoint of this study was set to be the diagnosis of ED (ICD-9-CM code 607.84 and 302.72) that recorded at least one inpatient or three outpatients [55], withdrawal, or end of follow-up. To ensure the reliability and accuracy of the diagnoses, the diagnoses had to be assigned by urologists for three or more ambulatory visits or at least one admission within a year. In a previous study, ICD-9 codes for ED have been validated, and corresponded to an International Index of Erectile Function score of less than 8 to define severe ED; using ICD-9 codes to enroll cases, the sensitivity was 80%, and the positive predictive value was 88.9% [56]. 

To eliminate potential bias, we adjusted for demographic variables, pre-existing co-morbidities and baseline co-medication, including hypertension (ICD-9-CM codes 401–405), diabetes mellitus (ICD-9-CM code 250), hyperlipidemia (ICD-9-CM code 272), stroke (ICD-9-CM code 430–438), coronary artery disease (ICD-9-CM code 410–414), chronic kidney disease (CKD) (ICD-9-CM code 585), chronic obstructive pulmonary disease (COPD) (ICD-9-CM codes 491,492, and 496), alcohol-related illness (ICD-9-CM codes 291, 303, 305, 571.0, 571.1, 571.2, 571.3, 790.3, A215, and V11.3), human immunodeficiency virus (HIV) infection (ICD-9-CM codes 042–044, 795.8, and V08), autoimmune disease (AID) including systemic lupus erythematosus (SLE) (ICD-9-CM code 710.0), rheumatoid arthritis (ICD-9-CM code 714.0), Sjögren’s syndrome (ICD-9-CM code 710.2), and multiple sclerosis (ICD-9-CM code 340). Other metrics that can affect ED were also adjusted in the analysis, including spinal cord injury pelvic fracture, burn of genitalia, foreign body entering through an orifice, injury to nerves and spinal cord, and poisoning. Information on comorbidities was obtained by tracing back 2 years before the index date about records of inpatients and ambulatory medical care in the database.

The medication confounders analyzed in this study consist of alpha-blockers, beta-blockers, diuretics, and calcium channel blockers (CCBs). Drug use was defined as the use of a drug for more than three prescriptions (with each prescription for 28 days) from outpatient visits prior to the index date.

### 2.4. Negative Control Exposure

A negative control exposure was designated to examine potential external confounders [57,58]. One can conduct an analysis with an alternative exposure that is not expected to be associated with the outcome of interest. In this study, Nontyphoidal *Salmonella* infection (NTS) was selected as the negative control exposure for its potential association with ischemic events [59,60]. However, there is no association between NTS and ED from a comprehensive review of the literature. We hypothesized that any association between NTS infection and ED implies possible unmeasured confounders. Accordingly, patients diagnosed with NTS infection (ICD-9-CM code 003.5) in at least three outpatient claims or one hospitalization from 2001 through 2012 and propensity score (PS)-matched non-salmonellosis controls were compared.

### 2.5. Statistical Analysis

Chi-squared tests were applied to compare the distribution of age, sex, and baseline co-morbidities between the HPV and non-HPV groups. The age mean was compared using Student’s *t*-test. The incidence density of ED per 1000 person-years was calculated in both groups. Multivariable Cox proportional hazards regression models were utilized to estimate the crude hazard ratios (cHRs), adjusted HRs (aHRs), and 95% confidence intervals (CIs) of ED for the HPV group versus the non-HPV group. The covariates adopted in the multivariate models encompassed sex, age, co-morbidities, and co-medications listed in Table 1. The Kaplan–Meier method was used to plot the cumulative incidence curves of ED for the HPV and non-HPV groups. Differences between the two groups were evaluated by the log-rank test. The incidence rates of ED were estimated by dividing the number of events by follow-up person-years for both groups. Two statistical models were fitted to evaluate the effect of HPV on the risk of ED (Table 2). In model 1, we examined the temporal relationship between HPV exposure and the risk of developing ED for both groups, adjusted for age, co-morbidities, and co-medications at baseline. Model 2 was adjusted for all covariates in model 1, and further adjusted for relevant operations associated with subsequent ED to minimize confounding, (e.g., foreign body entering through an orifice, low anterior resection, and abdominoperineal resection) [61,62,63]. Subgroup analyses were conducted to reveal the potential interaction effect of sex, age, and follow-up between HPV infection and subsequent ED development. All data analyses were performed with SAS (version 9.4; SAS Institute, Inc., Carey, NC, USA). The statistical significance level was set at a two-tailed *p*-value of <0.05.

### 2.6. Sensitivity Analyses

We performed a sensitivity analysis excluding patients with cancer and stroke at the index date to examine whether our findings remain constant under different assumptions.

## 3. Results

We enrolled 13,296 patients newly diagnosed with an HPV infection from 2000 to 2012 and 53,184 non-HPV controls in this study. The baseline characteristics of patients with and without an HPV infection were not significantly different, as all standard mean differences between the groups were <0.1 (Table 1). The mean ages of participants in the HPV and non-HPV groups were 39.4 (16.3) and 39.0 (15.7) years after PS matching, respectively.

In Table 2, model 1 and model 2 both show that patients with a history of HPV infection had a higher risk of developing ED (model 1, aHR, 1.64; 95% CI 1.38–1.95 and model 2, aHR, 1.63; 95% CI 1.37–1.94; *p* < 0.001). Table 2 also shows a sensitivity analysis, in which participants with severe comorbidities (cancer, stroke, and autoimmune diseases) that can cause (not just a risk factor for) ED were excluded. The results show consistent findings (aHR, 1.59; 95% CI 1.30–1.95; *p* < 0.001; details in Appendix A).

Negative control analysis showed that there was no association between NTS and subsequent ED with aHR, 0.69; 95% CI 0.31–1.53 (detail in Appendix A). This measurement revealed that there is minimal uncontrolled confounding from the point of view.

The finding that HPV infection is associated with higher ED risks was shown in the survival analysis, in which the cumulative incidence of ED for the HPV cohort was higher than that for the non-HPV cohort (log-rank test, *p* < 0.001) (Figure 1).

Table 3 shows that ED occurred significantly at around 40 years in this study and this finding was as consistent as the epidemiologic report [64]. Compared with individuals aged less than 30 years (this age group as reference), those aged 31–40 years had a higher risk of reporting ED (aHR, 1.89; 95% CI 1.40–2.56); for those aged 41–50 years, the aHR was 3.17 (95% CI 2.39–4.20). In Table 3, higher risks of ED were also identified among patients with hyperlipidemia (aHR, 1.63; 95% CI 1.34–1.97), alcohol-related illness (aHR, 1.43; 95% CI 1.01–2.05), AID (aHR, 1.66; 95% CI 1.09–2.53) and use of alpha- and beta-blockers (aHR, 4.07; 95% CI 3.33–4.97 and 1.30; 95% CI 1.03–1.64, respectively).

Table 4 shows the Cox proportional hazards regression models for comparing HPV and non-HPV patients in subgroup analyses. In the age subgroup of 31–40 years, compared with the age-matched non-HPV participants, those with HPV infection had a higher risk of developing ED (aHR, 1.65; 95% CI 1.05–2.60); in the age subgroup of 41–50 years, compared with the age-matched non-HPV participants, those with HPV infection had a higher risk of having ED (aHR, 1.63; 95% CI 1.17–2.27). The *p*-value for the interaction of age was not significant (*p =* 0.4). In comorbidities-subgroup analysis, among participants with diabetes, there was a positive association between HPV infection with ED (aHR, 2.39; 95% CI 1.63–3.51; *p* for interaction 0.05). Such positive association was also found among patients with stroke (aHR, 4.82; 95% CI 2.34–9.92), and COPD (aHR, 2.51; 95% CI 1.69–3.74) with *p* values for interaction 0.001 and 0.03, respectively.

In Table 5, subgroup analysis stratified by follow-up duration demonstrated that the risk of ED among HPV-infected patients was significantly higher in general after HPV infection. The risk of ED onset at <1, 1–5, and >5 years after HPV infection was 1.64 (95% CI = 1.38, 1.95), 1.48 (95% CI = 1.22, 1.80), and 1.7 (95% CI = 1.24, 2.34), respectively.

## 4. Discussion

In this large-scale population-based cohort study with up to 14 years of follow-up, we reported that individuals with HPV infection had a higher risk of developing ED. We selected symptomatic HPV patients as the study population because both inpatients and outpatients with symptoms are rarely studied. The subgroup analyses indicated that HPV-infected patients had a significantly higher risk of ED if they had a stroke or COPD. That is, COPD and stroke acted as modifiers for the relationship between HPV infection and ED. COPD is a consequence of tobacco use which is a well-known risk for ED. A time-to-event analysis indicated a positive association between HPV infection and ED in all the follow-up periods, which may be due to the characteristic of recurrent HPV infection [65]. However, the effect of persistent or recurrent HPV infection on the risk of ED has not been explored in this study. These findings are consistent in the sensitivity analysis and negative control analysis.

Previous studies on HPV infection have reported that women of reproductive age with positive HPV tests for cervical screening have poorer sexual function [66]. On the contrary, HPV infections among males have been investigated less, even though HPV DNA has been proven to lie not only in the perianal region and external genitalia, including the penis foreskin, scrotum, and glans penis, but also in the urethra, ductus deferens, epididymis, or testis [67]. For instance, HPV may infect the semen, contributing to male infertility [68]. This is due to decreased sperm progressive motility and the normal morphology rate caused by HPV infection [68,69]. Moreover, higher risks of anxiety and depression have been observed in patients with genital warts [16]. Having HPV causes fear, feeling of stigmatization, and worry about disclosing the disease to partners, family, or friends [16]. Furthermore, the psychological impact of HPV infection includes a negative association between genital warts and Arizona Sexual Experience Scale (ASEX) score, with lower ASEX scores being associated with a decline in the Beck Anxiety Inventory (BAI) and Beck Depression Inventory (BDI) indices [16]. Since the common etiologies of psychogenic ED include performance anxiety, a strained relationship, lack of sexual arousability, and overt psychiatric disorders such as depression and schizophrenia [70], it is likely that the psychological impact of HPV infection may also lead to ED.

In this cohort study, we employed a nationwide population-based registry to provide epidemiological evidence on the risks of new-onset ED in patients with HPV infection. The effect of measurable confounders was minimized by PS matching [71,72,73,74,75]. Sensitivity and negative control analyses were conducted to manage residual confounding and to ascertain the impact of HPV infection on ED development.

However, there are several limitations to this study. First, the evaluation of HPV infection is challenging, as many infections might not be clinically recognized [76,77,78]. The selection bias resulting from a retrospective study design means that only patients with a symptomatic HPV infection were included. That is, whether patients with asymptomatic or self-resolving infection also have a higher risk of developing ED remains unknown. Second, due to inherent database limitations, this study lacked penile ultrasound data, other indicators that may affect erectile function such as the use of phosphodiesterase-5 (PDE5) inhibitors, and an erectile performance questionnaire; however, in a previous study, the ICD-9 code for ED has been validated with the International Index of Erectile Function score [56]. Third, as our study only included participants from one country of Asian origin, our findings may not be generalizable to other ethnic groups.

## 5. Conclusions

Findings of this cohort study suggest that ED is an overlooked long-term complication of HPV infections in males. Future studies on HPV infections in males are warranted to facilitate the diagnosis and management of HPV infections.

## Figures and Tables

**Figure 1 jpm-12-00699-f001:**
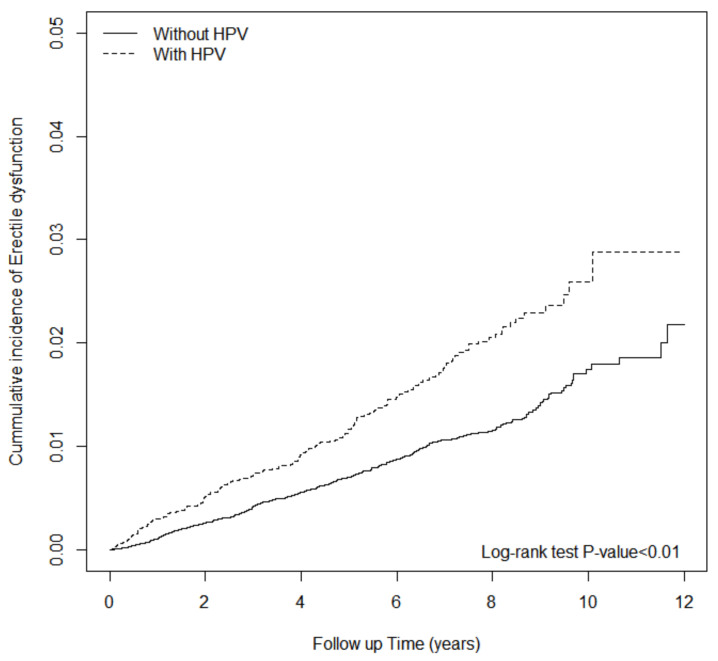
The cumulative incidence of erectile dysfunction in HPV-infected patients and non-HPV controls.

**Table 1 jpm-12-00699-t001:** The baseline characteristics in patients with and without HPV. infection.

Variables	Non-HPV		HPV		SMD
	(N = 53,184)		(N = 13,296)		
	*n*	%	*n*	%	
Age, year					
18–30	19,768	37%	4895	37%	0.007
31–40	10,967	21%	2710	20%	0.006
41–50	9873	19%	2399	18%	0.01
≥51	12,576	24%	3292	25%	0.03
mean, (SD)	39.0	(15.7)	39.4	(16.3)	0.03
Comorbidities					
hypertension	9821	18%	2492	19%	0.007
diabetes mellitus	4398	8.0%	1190	9.0%	0.02
hyperlipidemia	9419	18%	2458	18%	0.02
Stroke	2243	4.0%	654	5.0%	0.03
CAD	640	1.0%	217	2.0%	0.04
CKD	548	1.0%	187	1.0%	0.03
COPD	4987	9%	1351	10%	0.03
Alcohol-related illness	1838	3.0%	481	4.0%	0.009
HIV	80	0.0%	32	0.0%	0.02
AID	754	1.0%	238	2.0%	0.03
Medication					
α-blocker	7283	14%	1909	14%	0.02
β-blocker	3776	7%	1017	8%	0.02
CCB	9842	19%	2472	19%	0.002
diuretics	6470	12%	1673	13%	0.01

SMD: standard mean difference; CAD: cardiovascular disease; CKD: chronic kidney disease; COPD: chronic obstructive pulmonary disease; HIV: human immunodeficiency virus; AID: Autoimmune disease; CCB: calcium channel blocker.

**Table 2 jpm-12-00699-t002:** Hazard ratios for erectile dysfunction after diagnosis of HPV Infection.

	Hazard Ratio (95% CI)
**Primary analysis**	
Model 1 (adjusted comorbidities + comedications)	1.64 (1.38, 1.95) ***
Model 2 (adjusted comorbidities + comedications + relevant operation)	1.63 (1.37, 1.94) ***
**Sensitivity analyses**	
HPV excluding stroke, cancer, AID	1.59 (1.30, 1.95) ***
Alternative exposure (NTS)	0.69 (0.31, 1.53)

***: *p*-value < 0.001. Model 1: adjusted by age and hypertension, diabetes mellitus, hyperlipidemia, stroke, COPD, alcohol-related illness, AID (SLE, RA, SS), and all medication (a-blocker, b-blocker, calcium channel blocker, diuretics); Use of medication was defined as the prescription for at least three outpatients visit before the index date. Model 2: adjusted covariates in model 1 + relevant operation (foreign body entering through orifice and LAR: Low anterior resection; APR: Abdominoperineal resection).

**Table 3 jpm-12-00699-t003:** The association of explanatory variables and the risk of erectile dysfunction.

Variables	Erectile Dysfunction						(95% CI)
	n	PY	IR	cHR	(95% CI)	aHR ^ꝉ^	
non-HPV	432	286,137	1.51	1.00	-	1.00	-
HPV	181	71,478	2.53	1.69	(1.42, 2.00) ***	1.63	(1.37, 1.94) ***
Age, year							
18–30	78	141,370	0.55	1.00	-	1.00	-
31–40	93	75,174	1.24	2.25	(1.67, 3.04) ***	1.89	(1.40, 2.56) ***
41–50	176	65,317	2.69	4.93	(3.78, 6.44) ***	3.17	(2.39, 4.2) ***
≥51	266	75,754	3.51	6.49	(5.04, 8.36) ***	2.49	(1.83, 3.39) ***
Comorbidities							
hypertension							
No	416	298,632	1.39	1.00	-	1.00	-
Yes	197	58,983	3.34	2.43	(2.05, 2.88) ***	0.78	(0.60, 1.00) *
diabetes mellitus							
No	501	331,883	1.51	1.00	-	1.00	-
Yes	112	25,732	4.35	2.92	(2.38, 3.59) ***	1.21	(0.96, 1.52)
hyperlipidemia							
No	375	298,695	1.26	1.00	-	1.00	-
Yes	238	58,919	4.04	3.25	(2.76, 3.82) ***	1.63	(1.34, 1.97) ***
Stroke							
No	581	344,885	1.68	1.00	-	1.00	-
Yes	32	12,729	2.51	1.52	(1.06, 2.17) *	0.60	(0.41, 0.86) **
CAD							
No	602	353,790	1.7	1.00	-		
Yes	11	3825	2.88	1.70	(0.94, 3.09)		
CKD							
No	606	354,713	1.71	1.00	-		
Yes	7	2902	2.41	1.45	(0.69, 3.05)		
COPD							
No	509	325,821	1.56	1.00	-	1.00	-
Yes	104	31,794	3.27	2.10	(1.70, 2.59) ***	1.03	(0.82, 1.29)
Alcohol-related illness							
No	580	347,685	1.67	1.00	-	1.00	-
Yes	33	9930	3.32	2.05	(1.44, 2.91) ***	1.43	(1.01, 2.05) *
HIV							
No	613	357,109	1.72				
Yes	0	506	0				
AID							
No	590	352,918	1.67	1.00	-	1.00	-
Yes	23	4696	4.9	2.95	(1.94, 4.47) ***	1.66	(1.09, 2.53) *
Spinal cord injury							
No	613	357,445	1.71				
Yes	0	170	0				
Pelvic fracture							
No	613	357,599	1.71				
Yes	0	16	0				
Burn of unspecified degree of genitalia							
No	613	357,605	1.71				
Yes	0	9	0				
Foreign body entering through orifice							
No	523	321,822	1.63	1.00	-	1.00	-
Yes	90	35,793	2.51	1.57	(1.25, 1.96) ***	1.20	(0.96, 1.50)
Injury to nerves and spinal cord							
No	604	354,341	1.7	1.00	-		
Yes	9	3273	2.75	1.64	(0.85, 3.17)		
Poisoning							
No	610	354,612	1.72	1.00	-		
Yes	3	3003	1	0.58	(0.19, 1.82)		
Medication							
a-blocker							
No	316	307,984	1.03	1.00	-	1.00	-
Yes	297	49,631	5.98	5.81	(4.96, 6.81) ***	4.07	(3.33, 4.97) ***
b-blocker							
No	502	330,756	1.52	1.00	-	1.00	-
Yes	111	26,859	4.13	2.70	(2.19, 3.31) ***	1.30	(1.03, 1.64) *
CCB							
No	388	291,236	1.33	1.00	-	1.00	-
Yes	225	66,378	3.39	2.53	(2.15, 2.98) ***	1.02	(0.80, 1.30)
diuretics							
No	474	314,505	1.51	1.00	-	1.00	-
Yes	139	43,110	3.22	2.12	(1.76, 2.57) ***	0.74	(0.59, 0.93) *
Treatment							
Radical cystectomy							
No	613	357,555	1.71				
Yes	0	60	0				
LAR + APR							
No	610	357,350	1.71	1.00	-	1.00	-
Yes	3	265	11.3	6.56	(2.11, 20.41) **	2.70	(0.86, 8.43)

*: *p*-value < 0.05; **: *p*-value < 0.01; ***: *p*-value < 0.001; PY: person-years; IR: incidence rate per 1000 person-years; cHR: crude hazard ratio; aHR: adjusted hazard ratio. CAD: cardiovascular disease; CKD: chronic kidney disease; COPD: chronic obstructive pulmonary disease; HIV: human immunodeficiency virus; AID: Autoimmune disease; CCB: calcium channel blocker; LAR: Low anterior resection; APR: Abdominoperineal resection. ^ꝉ^: adjusted by age and hypertension, diabetes mellitus, hyperlipidemia, stroke, COPD, alcohol-related illness, AID, and all medication.

**Table 4 jpm-12-00699-t004:** Subgroup analyses for the effect of HPV infection on ED risks under different stratification.

Variables	Non-HPV			HPV							*p* for Interaction
	n	PY	IR	n	PY	IR	cHR	(95% CI)	aHR ^ꝉ^	(95% CI)	
Age, year											0.40
18–30	61	113,415	0.54	17	27,955	0.61	1.14	(0.66, 1.94)	1.14	(0.67, 1.95)	
31–40	67	60,335	1.11	26	14,839	1.75	1.61	(1.03, 2.54) *	1.65	(1.05, 2.6) *	
41–50	127	52,646	2.41	49	12,671	3.87	1.61	(1.16, 2.24) **	1.63	(1.17, 2.27) **	
≥51	177	59,740	2.96	89	16,013	5.56	1.87	(1.45, 2.42) ***	1.85	(1.43, 2.39) ***	
Comorbidities											
hypertension											0.10
No	303	239,303	1.27	113	59,329	1.9	1.51	(1.22, 1.87) ***	1.5	(1.21, 1.87) ***	
Yes	129	46,833	2.75	68	12,150	5.6	2.05	(1.53, 2.76) ***	1.99	(1.48, 2.68) ***	
diabetes mellitus											0.05
No	364	266,014	1.37	137	65,869	2.08	1.52	(1.25, 1.86) ***	1.5	(1.23, 1.82) ***	
Yes	68	20,122	3.38	44	5610	7.84	2.38	(1.63, 3.49) ***	2.39	(1.63, 3.51) ***	
hyperlipidemia											0.89
No	265	239,430	1.11	110	59,265	1.86	1.68	(1.34, 2.1) ***	1.69	(1.36, 2.12) ***	
Yes	167	46,706	3.58	71	12,213	5.81	1.65	(1.25, 2.18) ***	1.58	(1.19, 2.09) **	
Stroke											0.001
No	420	276,453	1.52	161	68,433	2.35	1.56	(1.3, 1.87) ***	1.53	(1.28, 1.84) ***	
Yes	12	9684	1.24	20	3046	6.57	5.18	(2.53, 10.6) ***	4.82	(2.34, 9.92) ***	
CAD											0.46
No	424	283,349	1.5	178	70,441	2.53	1.7	(1.42, 2.02) ***	1.67	(1.4, 1.99) ***	
Yes	8	2787	2.87	3	1037	2.89	1.07	(0.28, 4.07)	1.3	(0.32, 5.23)	
CKD											0.27
No	429	283,990	1.51	177	70,723	2.5	1.66	(1.4, 1.98) ***	1.63	(1.36, 1.94) ***	
Yes	3	2147	1.4	4	755	5.3	3.64	(0.81, 16.27)	4.99	(0.94, 26.6)	
COPD											0.03
No	370	261,178	1.42	139	64,642	2.15	1.52	(1.25, 1.85) ***	1.5	(1.23, 1.82) ***	
Yes	62	24,958	2.48	42	6836	6.14	2.56	(1.73, 3.81) ***	2.51	(1.69, 3.74) ***	
Alcohol-related illness											0.17
No	413	278,340	1.48	167	69,345	2.41	1.63	(1.36, 1.95) ***	1.6	(1.34, 1.92) ***	
Yes	19	7796	2.44	14	2134	6.56	2.68	(1.34, 5.34) **	2.52	(1.25, 5.07) **	
HIV											1.00
No	432	285,790	1.51	181	71,319	2.54	1.69	(1.42, 2.01) ***	1.64	(1.38, 1.96) ***	
Yes	0	347	0	0	159	0					
AID											0.17
No	420	282,571	1.49	170	70,347	2.42	1.63	(1.37, 1.95) ***	1.62	(1.35, 1.93) ***	
Yes	12	3566	3.37	11	1131	9.73	2.85	(1.26, 6.46) *	2.44	(1.07, 5.6) *	
Medication											
a-blocker											0.47
No	228	246,731	0.92	88	61,253	1.44	1.56	(1.22, 1.99) ***	1.55	(1.21, 1.98) ***	
Yes	204	39,405	5.18	93	10,226	9.09	1.76	(1.37, 2.25) ***	1.75	(1.37, 2.24) ***	
b-blocker											0.82
No	355	264,742	1.34	147	66,014	2.23	1.66	(1.37, 2.02) ***	1.63	(1.35, 1.98) ***	
Yes	77	21,394	3.6	34	5464	6.22	1.75	(1.16, 2.62) **	1.7	(1.13, 2.56) *	
CCB											0.07
No	283	232,996	1.21	105	58,240	1.8	1.49	(1.19, 1.86) ***	1.45	(1.16, 1.82) **	
Yes	149	53,140	2.8	76	13,238	5.74	2.06	(1.56, 2.72) ***	2.02	(1.53, 2.67) ***	
Diuretics											0.25
No	340	251,848	1.35	134	62,657	2.14	1.58	(1.3, 1.93) ***	1.54	(1.26, 1.88) ***	
Yes	92	34,289	2.68	47	8821	5.33	2.06	(1.45, 2.93) ***	2.07	(1.45, 2.96) ***	

*: *p*-value < 0.05; **: *p*-value < 0.01; ***: *p*-value < 0.001; PY: person-years; IR: incidence rate per 1000 person-years; cHR: crude hazard ratio; aHR: adjusted hazard ratio. CAD: cardiovascular disease; CKD: chronic kidney disease; COPD: chronic obstruction pulmonary disease; HIV: human immunodeficiency virus; AID: Autoimmune disease; CCB: calcium channel blocker; ^ꝉ^: adjusted by age and hypertension, diabetes mellitus, hyperlipidemia, stroke, COPD, alcohol-related illness, AID and all medication.

**Table 5 jpm-12-00699-t005:** Effect of HPV infection on ED development, stratified by follow-up duration to ED onset.

	Non-HPV			HPV						
Follow-Up, Year	*n*	PY	IR	N	PY	IR	cHR	(95% CI)	aHR ^ꝉ^	(95% CI)
<1	58	52,691	0.11	40	13,210	0.30	1.67	(1.41, 1.99) ***	1.64	(1.38, 1.95) ***
1–5	238	262,705	0.09	86	65,757	0.13	1.51	(1.24, 1.83) ***	1.48	(1.22, 1.80) ***
>5	136	212,530	0.06	55	53,049	0.10	1.67	(1.22, 2.29) **	1.7	(1.24, 2.34) ***

**: *p*-value < 0.01; ***: *p*-value < 0.001; PY: person-years; IR: incidence rate (per 1000 person-years): cHR: crude hazard ratio; aHR: adjusted hazard ratio. ^ꝉ^: adjusted by age and hypertension, diabetes mellitus, hyperlipidemia, stroke, COPD, alcohol-related illness, AID and all medication.

## Data Availability

The data used in this study was held by the Taiwan Ministry of Health and Welfare, and thus was not publicly available. The data can be accessed by submitting an application to the Ministry of Health and Welfare for access. (Taiwan Ministry of Health and Welfare, Address: No. 488, Sec. 6, Zhongxiao E. Rd., Nangang Dist., Taipei City 115, Taiwan (R.O.C.). Phone: +886-2-8590-6848).

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
