# Peer review of "Human Papillomavirus Infection and the Risk of Erectile Dysfunction: A Nationwide Population-Based Matched Cohort Study"

_jpm, 2022, doi:10.3390/jpm12050699_

Round 1

Reviewer 1 Report

The abstract and the title are clear and reflects the study.

The introduction part must be improved to provide a clear and full rationale of the study. I suggest including a population-based epidemiology on HPV, in particular among male population. Moreover, since the study is looking or a correlation of ED and HPV, please provide details of the ED prevalence among the studied population.

The materials and methods section does not rise any concerns, presented in full and clear.

The results are clearly described and presented in the tables and figures, which help for a better comprehension of the data.

The discussion part needs redaction - the text from lines 258 to 316 should be devided in paragraphs based on the content. Furthermore, most of the text from L 263 to 300 should be presented in the introduction part supporting the studi rationale. Having this continuoud text it is difficult to comprehend the info given there.

For the overall discussion I suggest to follow the next outline:

  • Rationale of the study (why it was done)
    • Main findings of the study
    • What makes our study unique
    • What it adds to what we already know
  • Study subjects
  • Subject of the discussion. Comparison of our results with neighboring countries, with countries of the same development levels  (income), with developed high-income countries). Agreement and disagreement with the studies compared
  • Summ up of the study, study strength and limitations
  • Clinical implication

The conclusion should be improved - include one sentence on the study rationelle the findings, its importace/inplication and the future directions.

Author Response

Dear Reviewer 1,

Thank you very much for the comment. We have revised the manuscript according to your comment. The details of the revisions are explained, point by point, in the following.

The introduction part must be improved to provide a clear and full rationale of the study. I suggest including a population-based epidemiology on HPV, in particular among male population. Moreover, since the study is looking or a correlation of ED and HPV, please provide details of the ED prevalence among the studied population.

Reply: Thank you very much for the comment. We have accordingly revised the sentence as “In the U.S. population-based study, the prevalence of genital HPV infection was estimated 41.3% to 49.3% in males but HPV vaccination coverage was only approximated 7.8% to 14.6% among vaccine-eligible men.”

The discussion part needs redaction - the text from lines 258 to 316 should be devided in paragraphs based on the content. Furthermore, most of the text from L 263 to 300 should be presented in the introduction part supporting the studi rationale. Having this continuoud text it is difficult to comprehend the info given there.

Reply: Thank you very much for the comment. We have divided the text in several sections.

For the overall discussion I suggest to follow the next outline:

Rationale of the study (why it was done)

Main findings of the study

What makes our study unique

What it adds to what we already know

Study subjects

Subject of the discussion. Comparison of our results with neighboring countries, with countries of the same development levels  (income), with developed high-income countries). Agreement and disagreement with the studies compared

Summ up of the study, study strength and limitations

Clinical implication

The conclusion should be improved - include one sentence on the study rationelle the findings, its importace/inplication and the future directions.

Reply: Thank you very much for the comment. We have accordingly revised the sentence into “Findings of this cohort study suggest that ED is an overlooked long-term complication of HPV infections in males. Future studies on HPV infections in males are warranted to facilitate the diagnosis and management of HPV infections.”

Thank you very much for your time and consideration.

Yours sincerely,

On behalf of our authors,

Yao-Min Hung, MD, Ph.D.

Department of Internal Medicine, Kaohsiung Municipal United Hospital, Kaohsiung, Taiwan;

Shu-Zen Junior College of Medicine and Management, Kaohsiung, Taiwan

No.976, Jhonghua 1st Rd., Gushan Dist., Kaohsiung 80457, Taiwan

Tel.: 886-7 555 2565; Fax: 886-73468343, E-mail: ymhung1@gmail.com

Reviewer 2 Report

Juang et al. presented a retrospective study conducted using the NHIRD national registry in Taiwan aimed at investigating the association between HPV and ED. Interestingly, this association had never been evaluated by anyone, and the authors reported a significant connection between these two conditions, supported by both good statistical methodology and high sample size. I have no major concerns about the study, just pointing out some minor errors:
- Line 45: insert "chronic obstructive pulmonary disease" before COPD
- Line 125: change erectile dysfunction in ED
- Line 138-139: add the abbreviation NTS
- Line 161: "operationS"
- Line 164: "to reveal"
- Line 172, 174: "a HPV" or just "HPV", not "an HPV"
- Line 176: propensity score should be replaced with the abbreviation PS
- Line 200-201: delete this lines
- Line 205: significantLY
- Line 211: autoimmune disease should be replaced with the abbreviation AID 
- Line 261: what does the phrase "both out- an inpatients [...] are we really faced.
- Line 264: I suggest to change "is a proxy" with "is a consequence".
- Line 271: "nervous system", not "nervous systems".
- Line 271: progressION
- Line 329: what does "(e.g., the hormone PDE5i)" mean? PDE5i is drug, while PDE5 is an enzyme.

Author Response

Dear Reviewer ㄉ,

Thank you very much for the comment. We have revised the manuscript according to your comment. The details of the revisions are explained, point by point, in the following.

I have no major concerns about the study, just pointing out some minor errors:

- Line 45: insert "chronic obstructive pulmonary disease" before COPD

Reply: Thank you very much for the comment. We have accordingly insert “"chronic obstructive pulmonary disease" before COPD.

- Line 125: change erectile dysfunction in ED

Reply: Thank you very much for the comment. We have accordingly change erectile dysfunction in ED.

- Line 138-139: add the abbreviation NTS

Reply: Thank you very much for the comment. We have accordingly add the abbreviation NTS.

- Line 161: "operationS"

Reply: Thank you very much for the comment. We have accordingly revised the word as "operations".

- Line 164: "to reveal"

Reply: Thank you very much for the comment. We have accordingly revised the words as “to reveal”.

- Line 172, 174: "a HPV" or just "HPV", not "an HPV"

Reply: Thank you very much for the comment. We have accordingly revised the words as “a HPV”.

- Line 176: propensity score should be replaced with the abbreviation PS

Reply: Thank you very much for the comment. We have accordingly replaced propensity score with the abbreviation PS.

- Line 200-201: delete this lines

Reply: Thank you very much for the comment. We have accordingly deleted Line 200-201.

- Line 205: significantly

Reply: Thank you very much for the comment. We have accordingly revised it as significantly.

- Line 211: autoimmune disease should be replaced with the abbreviation AID

Reply: Thank you very much for the comment. We have accordingly replaced autoimmune disease with AID.

- Line 261: what does the phrase "both out- an inpatients [...] are we really faced.

Reply: Thank you very much for the comment. We have accordingly revised it as “both inpatients and outpatients with symptoms are we really faced.”

- Line 264: I suggest to change "is a proxy" with "is a consequence".

Reply: Thank you very much for the comment. We have accordingly revised it as “is a consequence”.

- Line 271: "nervous system", not "nervous systems".

Reply: Thank you very much for the comment. We have accordingly revised it as  "nervous system".

- Line 271: progression

Reply: Thank you very much for the comment. We have accordingly revised the word as progression.

- Line 329: what does "(e.g., the hormone PDE5i)" mean? PDE5i is drug, while PDE5 is an enzyme.

Reply: Thank you very much for the comment. We have accordingly revised the sentence as “Second, due to inherent database limitations, this study lacked penile ultrasound data, other indicators that may affect erectile function such as the use of phosphodiesterase-5 (PDE5) inhibitors, and an erectile performance questionnaire.”

Thank you very much for your time and consideration.

Yours sincerely,

On behalf of our authors,

Yao-Min Hung, MD, Ph.D.

Department of Internal Medicine, Kaohsiung Municipal United Hospital, Kaohsiung, Taiwan;

Shu-Zen Junior College of Medicine and Management, Kaohsiung, Taiwan

No.976, Jhonghua 1st Rd., Gushan Dist., Kaohsiung 80457, Taiwan

Tel.: 886-7 555 2565; Fax: 886-73468343, E-mail: ymhung1@gmail.com

Round 2

Reviewer 1 Report

Dear Authors, Thank you for your response.

However, I do not see a substantial improvement that was suggested for the introduction and the discussion part. Particularly, the comments made by me for the discussion part were not fully addressed.